# Malicious Office Macro Detection: Combined Features with Obfuscation and Suspicious Keywords

**Xiang Chen, Wenbo Wang and Weitao Han ***

Institute of Information Technology, PLA Strategic Support Force Information Engineering University, Zhengzhou 450001, China; chenxndsc@163.com (X.C.); wbwangedu@outlook.com (W.W.)
* Correspondence: weitaohanchn@163.com

**Abstract:** Microsoft has implemented several measures to defend against macro viruses, including the use of the Antimalware Scan Interface (AMSI) and automatic macro blocking. Nevertheless, evidence shows that threat actors have found ways to bypass these mechanisms. As a result, phishing emails continue to utilize malicious macros as their primary attack method. In this paper, we analyze 77 obfuscation features from the attacker's perspective and extract 46 suspicious keywords in macros. We first combine the aforementioned two types of features to train machine learning models on a public dataset. Then, we conduct the same experiment on a self-constructed dataset consisting of newly discovered samples, in order to verify if our proposed method can identify previously unseen malicious macros. Experimental results demonstrate that, compared to existing methods, our proposed method has a higher detection rate and better consistency. Furthermore, ensemble multi-classifiers with distinct feature selection can further enhance the detection performance.

**Keywords:** malicious document; VBA; macro; obfuscation; suspicious keywords; machine learning

## 1. Introduction

Dealing with Office documents has become an integral part of work in the digital era. Unfortunately, it has become a common tactic for attackers to carry out cyberattacks using malicious documents. A typical approach is that an attacker crafts a malicious document that exploits a vulnerability or contains a malicious macro. They then add the document as an email attachment and send the email to the target. Since Office documents are often perceived as non-threatening, there is a high probability that the target will open the malicious document. However, continuous improvements in platforms and applications have significantly mitigated malicious activities, while attacks based on social engineering and macro abuse have surged in the past decade [1]. To counteract this threat, Microsoft introduced the Antimalware Scan Interface (AMSI) in 2018. This interface enables antivirus security solutions to scan macros and other scripts at runtime. It has also been integrated into the new version of the Office 365 client application. Last year, Microsoft Office implemented a proactive measure by blocking macros from the internet. This action resulted in a 66% reduction in macro-based attacks over an eight-month period. However, as attackers refine their techniques to bypass this mechanism, macro-based Office documents still pose a security threat. Security researchers have observed that attackers have been increasingly utilizing alternative document formats, such as Publisher files and OpenDocument text (.odt) files, to conduct phishing attacks. These formats may still execute macros, but they can bypass the latest security controls. Despite this, several Advanced Persistent Threat (APT) groups, such as Kimsuky, Donot, and SideCopy, as well as cybercriminals like Emotet, continue to rely on macro-based Office documents to launch attacks and distribute their malicious payloads.

Current research focuses on detecting malicious macros by examining macro obfuscation and suspicious keywords, such as specific strings and functions. Both methods have limitations. Obfuscation detection is unable to distinguish between the obfuscation of

benign or malicious macros, resulting in numerous false positives in detection [2]. The latter detection method relies on the knowledge of security experts regarding known suspicious keywords and has been proven to be incapable of detecting new attack patterns.

In fact, the aforementioned approaches appear to complement each other. Obfuscation is often necessary for malicious macros, regardless of the attack techniques used. Therefore, obfuscation detection can be employed to identify unknown malicious macros. On the other hand, detection based on suspicious keywords can capture known malicious characteristics of macros. However, the two types of features mentioned above have never been combined to identify malicious macros in previous research.

Based on the above observations, this paper conducts an objective analysis by extracting features of obfuscation and suspicious keywords and integrating them to detect malicious macros. First, 123 features are thoroughly extracted from obfuscation and suspicious keywords. Second, four widely used machine learning models are trained to detect malicious macros. Our proposed method is evaluated on two datasets: Dataset1, a public dataset, contains 2939 benign samples and 13,734 malicious samples. Dataset2 consists of 2885 new samples from Virustotal, all of which were reported after the publication date of Dataset1. The experiments conducted on Dataset1 reveal that the adjusted model with combined features achieves the highest F1-score of 0.997. Employing this classifier to detect malicious samples from Dataset2 achieves a detection rate of 95.3%. Comparing the detection of macros using a single type of feature, the classifier with combined features achieves better performance. Further analysis shows that conducting ensemble training could further improve the detection performance.

This paper contributes in the following ways:

- We extract a concise and consistent set of macro obfuscation features from the perspective of adversarial attacks.
- We analyze the drawbacks of using obfuscated features or suspicious keywords in the detection of malicious macros, and for the first time, we combine these two types of features in machine learning models to detect malicious macros. Our approach has a very low rate of false positives and is capable of detecting unknown malicious macros.

The paper is organized as follows: Section 2 outlines the background and details the techniques for obfuscating MS Office macros. In Section 3, we present recent research on detecting malicious macros in Office documents. We present our approach in Section 4 and evaluate it in Section 5. Sections 6 and 7 include the discussion and conclusion of this paper.

## 2. Background

In this section, we will introduce the background knowledge related to VBA macros and focus on common obfuscation techniques used in macros.

### 2.1. Macro in Office Documents

MS Office macros, developed in the Visual Basic for Applications (VBA) programming language, have been widely applied in Microsoft Office documents such as Word, Excel, PowerPoint, and Outlook. Users can develop customized macro codes to interact with Office applications. These codes can be used to complete various tasks, such as manipulating data, formatting documents, creating charts, sending emails, etc. The automation of repetitive tasks is one of the main advantages of utilizing macros. For example, when users need to perform formatting actions multiple times in Microsoft Word, such as applying a specific font, setting paragraph alignment, or inserting headers and footers, they can use macros to record these actions once and execute them whenever necessary. This helps to save time and effort. Macros also help users accomplish complex tasks that may otherwise be challenging or time-consuming. For example, Microsoft Excel enables its users to create macros. These macros can analyze large datasets, perform calculations, generate reports, and automatically create charts when triggered by a specific event, such as workbook opening or button clicking. Figure 1 shows the VBA macros of two benign documents.

Figure 1a demonstrates how to modify the font format in a Word document, while Figure 1b shows a macro snippet that automatically sends an email with the content of the document.

```vba
                    Public Sub SaveNSend()
                    ...
                        Set otlApp = CreateObject("Outlook.Application")
                        Set otlNewMail = otlApp.CreateItem(olMailItem)
                        fname = TempFilePath & Filename & FileExt

                        With otlNewMail
                        .To = "sto.lgdr@sto.sc.gov"
                        .Subject = Filename
Sub ApplyFont()          .Body = County & " County" & Chr(10) _
    Selection.Font.Name = "Arial"       & "Entity Type: " & ReportingEntityType & Chr(10) _
    Selection.Font.Size = 12            & "Entity Name: " & ReportingEntityName & Chr(10) _
    Selection.Font.Bold = True          & "Fiscal Year Ending " & FiscalYear & Chr(10) _
End Sub                                 & "Submitted on " & TodaysDate
                        .Attachments.Add fname
                        .Send
                        End With
                    ...
                    End Sub
```

(**a**)                                             (**b**)

**Figure 1.** VBA macro examples. (**a**) A procedure of VBA macro used to change font format. (**b**) A snippet of VBA macro used to send an email.

While bringing convenience to users, macros are also exploited by attackers to carry out various malicious activities. As depicted in Figure 2, a straightforward yet malicious macro, comprising only four lines, can still execute a series of malicious actions. It obtains a running instance of Windows Explorer using the VBA function "GetObject". The method "ShellExecute" is then called to initiate PowerShell, which downloads and executes a malicious script. There are various techniques that can be employed to execute commands, as we will further detail in the following sections.

```vba
Sub Document_Open()
    Set obj = GetObject("new:C08AFD90-F2A1-11D1-8455-00A0C91F3880")
    obj.Document.Application.ShellExecute "powershell.exe", "IEX (New-Ob
ject Net.WebClient).DownloadString('http://192.168.131.127:8000/httptest1.ps1')",
    "C:\\Windows\\System32", Null, 0
End Sub
```

**Figure 2.** Example of malicious VBA macros.

### 2.2. Obfuscation of VBA Macro

Code obfuscation was proposed as a method to prevent reverse analysis. In legitimate situations, macros are typically obfuscated to protect the intellectual property of the source code. However, in the context of malicious macros, obfuscation is primarily used to hide the malicious commands, making it challenging for antivirus software to identify any suspicious elements. We list six commonly used techniques for obfuscating VBA macros:

- Random Obfuscation: Randomize the variable names or procedure names in a macro with randomly generated characters. Figure 3 provides an example of random obfuscation. Both the variable names and function parameters have been obfuscated.

```
Dim COHdDxBvIVN() As Byte: COHdDxBvIVN = oymlTquSfEJJz(gZOnGYtBrHWbfY)
Dim nMLloAyJokfWI As Long: nMLloAyJokfWI = UBound(COHdDxBvIVN)
```

**Figure 3.** Example of random obfuscation.

- Split Obfuscation: Long strings can be split into several shorter strings connected with "+" or "=". This operation is common in regular macros when a lengthy string needs to be concatenated with multiple sub-strings. Figure 4a demonstrates split obfuscation in a regular macro, while Figure 4b shows a snippet of a malicious macro utilizing split obfuscation for the parameter of the "GetObject" function.

```
If IsError(found) Then
    msg = msg & rowMsg & " " & cellAddress & " - Invalid, select from list." & vbCrLf
End If
```

```
lw = ""
Set obj = GetObject("new:C08A" + lw + "FD90-F" + lw + "2A1-11D1-845" + lw + "5-00A0C91F3880")
```

(**a**) (**b**)

**Figure 4.** Split obfuscation examples. (**a**) Example of split obfuscation from a benign macro. (**b**) Example of split obfuscation from a malicious macro.

- Encoding Obfuscation: Encoding obfuscation is similar to encryption and requires decoding operations to restore the original string before it can be used. For instance, we can encode a PowerShell command as a byte array or a Base64 string. PowerShell commands are often targeted for encoding obfuscation. Figure 5 demonstrates the parameter of the "GetObject" function being obfuscated using byte arrays and subsequently decoded with a self-defined decoding function called "FicTuJxzQrLu".

```
Set GswGsBPaLDzb = GetObject(FicTuJxzQrLu(Array(((16 Xor 1)+84),(3+110),(6 Xor 11),
(1 Xor 17),112,72),(0 Xor 0)) & FicTuJxzQrLu(Array((13+(50 Xor 172)),139,((14 Xor
21)+51),(29+(3 Xor 15)),(101+98),(223+(19 Xor 5)),(35+(18 Xor 55)),((2 Xor 11)+(13
 Xor 20)),((24 Xor 75)+64),((21 Xor 61)+73),148,((36 Xor 176)+(0 Xor 29)),89,(14
  Xor 197),(23 Xor 222),((18 Xor 14)+126),(74 Xor 170),((7 Xor 15)+(0 Xor 2)),(7
  Xor 23),80,(6+0),(3 Xor 12),(0 Xor 145),((14 Xor 158)+(0 Xor 13)),((35 Xor 20)
  +160), (106 Xor 220),(44+(20 Xor 49)),(66+90),92,(87+(58 Xor 184)),173,(15 Xor
  111),157,(63 Xor 113)),(5+(0 Xor 1)))))
```

**Figure 5.** Example of encoding obfuscation.

- Embedding Obfuscation: Embedding obfuscation conceals the target string in other parts of the document, such as document attributes, forms, or controls, and even within the content of the document. The macro can then load this content directly into a string variable. This obfuscation can be challenging to recognize because it is a common operation in non-obfuscated macros for processing content. Figure 6 provides an example of embedding obfuscation. The PowerShell command is loaded from a hidden shape within the document.

```
'Get the string in the document
content = Mid(ActiveDocument.Shapes.Item(1).AlternativeText, 6)
```

**Figure 6.** Example of embedding obfuscation.

- Call obfuscation: When dealing with sensitive object methods such as "Run", "ShellExecute", and "Create" in VBA macros, the "CallByName" method can be useful for executing a method by specifying its name as a string. By combining it with other forms of string obfuscation, the called method can be further obfuscated. The macro depicted in Figure 7b is the obfuscated version of the code shown in Figure 7a, where the object's called method is obfuscated.

```
Set o = CreateObject("Wscript.Shell")      Set o = CreateObject("Wscript.Shell")
o.Run "Calc.exe"                           methodname = Chr(114) + "un"
                                           CallByName o, methodname, VbMethod, "Calc.exe"
```

                 **(a)**　　　　　　　　　　　　　　　　　　　　　　　　**(b)**

**Figure 7.** Example of call obfuscation. (**a**) Example of a macro before call obfuscation. (**b**) Example of a macro after call obfuscation.

- Logical Obfuscation: The concept of logical obfuscation involves adding dummy code such as procedure calls and branch decisions, which complicates the logic of the macro. In certain cases, attackers conceal a few lines of malicious statements among large amounts of normal macro code, making it problematic to analyze malicious macros. This situation can be likened to searching for a needle in a haystack. Without understanding the program semantics, it is impossible to determine if the code is logically obfuscated.

## 3. Related Work

In recent years, many researchers have focused on detecting malicious JavaScript code in PDF documents through static analysis [3–7] or dynamic analysis [8–10]. However, there are still few studies on detecting malicious Office macros. Most of these research studies focus on machine-learning-based static detection, and some recent related works are described below.

Mimura et al. [11–13] utilized natural language processing (NLP) techniques to extract vocabularies from macros by treating them as text. They then employed word embedding, latent semantic indexing (LSI), and other methods to extract feature vectors for training machine learning models in order to detect malicious macros. The effectiveness of these models in detecting new samples over time was experimentally analyzed. Nir Nissim et al. [14] extracted structural features from DOCX documents and used machine learning to identify malicious DOCX documents. Utilizing active learning with the assistance of experts, the system achieved a 94.44% detection rate with a low false alarm rate of 0.19%. The main limitation of the research is that it only applies to DOCX documents. Kim et al. [2] investigated VBA macro obfuscation in Office documents. They categorized VBA macro obfuscation techniques into four types and introduced a feature set for effectively detecting obfuscation. However, they clarified that their study is aimed at detecting obfuscation, not malicious macros. While obfuscation techniques are implemented to protect intellectual property rights, it is imperative to differentiate between malicious and obfuscated macro detection. Failing to do so will result in an increased occurrence of false alarms when using obfuscation characteristics to detect malicious macros. Vasilios et al. [15] analyzed a significant number of Office documents that contained VBA macros. They used 40 suspicious keywords as features to identify malicious macros. The machine learning model, random forest, achieved an accuracy of 97.5%. However, it appears that these features do not adequately cover certain malicious macros that utilize new keywords, let alone unknown ones in the future.

Other than VBA source code, Bearden et al. [16] concentrated on the compiled version of VBA source code, known as p-code, and introduced a machine-learning-based approach for identifying malicious macro documents. The detection is performed by extracting n-gram features from p-code sequences, and the classification accuracy is 96.3% on 158 experimental samples. Simon et al. [17] adopted 2-g features and obfuscation features from p-code to identify malicious documents. They also increased the number of document samples to 20,196. The experiments showed an accuracy of 98.8 percent. As XL4 macros in Excel have become significant attack vectors, Ruaro et al. [18] introduced SYMBEXCEL, which utilizes symbolic execution to automatically de-obfuscate and analyze these macros. This enables users to extract Indicators of Compromise (IoCs) and other vital forensic information reliably.

## 4. Machine Learning Method with Combined Features

Usually, a machine-learning-based model consists of five typical steps: data collection, pre-processing, feature extraction, model training, and evaluation, as shown in Figure 8. In this section, we will describe the macro sample collection, pre-processing, and the extraction of detection features. The last two steps will be specified in the next section.

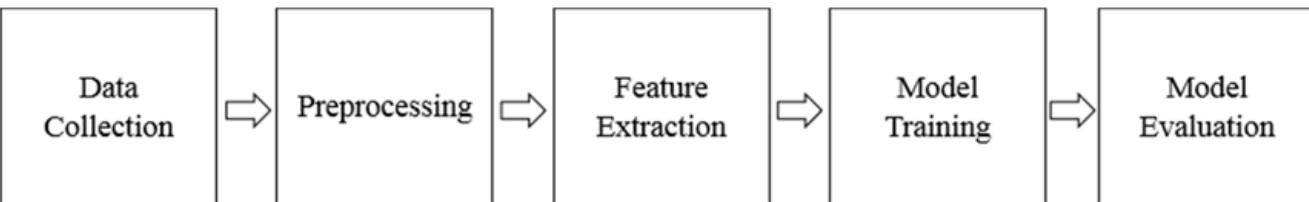

**Figure 8.** Typical steps of a machine-learning-based model.

### 4.1. Data Collection and Pre-Processing

To compare with previous studies, we use the same dataset from Vasilios [19], comprising 2968 benign Office documents and 15,570 malicious Office documents, all of which are macro-based. The benign samples are collected from 726 governmental and 1010 educational sites across various countries and institutions. Meanwhile, the malicious samples originate from three well-known repositories: AppAny, Virusign, and Malshare.

In the pre-processing phase, we extract the VBA macros from the available samples and remove those that do not meet our requirements, as shown in the subsequent section. Subsequently, we label the remaining data for training and testing. To extract macros from Office documents, we utilize Oletools, an open-source Python-based tool, which is also used for extracting suspicious keywords.

- Malicious samples disarmed

After conducting a preliminary analysis of the macros in this dataset, we discovered that several malicious macros contain the message "Macro code was removed by Symantec Disarm". This suggests that the malicious part of the macros has been removed, and therefore it may not be appropriate to classify them as malicious. To confirm this, the malicious samples were rescanned by ClamAV antivirus software (Software version 0.103.10, Virus database updated at 16 September 2021). After scanning, 14,064 samples were identified as malicious, and the remaining 1506 samples were removed from the dataset of malicious samples.

- Excel 4.0 macro samples

As Excel 4.0 macros differ significantly from VBA macros, detecting Excel 4.0 macros is beyond the scope of our method. Therefore, we excluded pure Excel 4.0 macros to achieve more accurate results and retained samples that contained both VBA and Excel 4.0 macros. We acquired 171 samples with malicious Excel 4.0 macros and found no Excel 4.0 macros in benign samples.

- Samples unable to parse with Oletools

It is noteworthy that certain malicious samples cannot be parsed successfully using Oletools. These samples appear to be intentionally designed to exploit the weaknesses of Oletools and evade detection. This highlights the strategic maneuvers and countermeasures in the ongoing battle between attackers and defenders. A total of 155 malicious samples could not be parsed by Oletools. However, the tool is able to successfully parse all benign samples.

- Samples without macros

We still find some samples without macros and discard them.

After pre-processing, our final dataset contains 2939 benign samples and 13,734 malicious samples that include VBA macros. Table 1 presents the breakdown of document types

and quantities of both benign and malicious samples. As shown in Table 1, Excel documents are the main format containing benign macros, whereas Word documents are preferred by attackers with malicious intentions. We believe this is because Word documents are more appealing than Excel documents in phishing emails. In addition, there are various other types of malicious documents, such as Office Publisher files, encrypted documents with CDFV2.

**Table 1.** Details of purified dataset from Vasilios.

| Label | Benign | Malicious |
|---|---|---|
| number | 2939 | 13,734 |
| Details of document type | DOC 644, DOCX 26, XLS 1607, XLSX 662 | DOC 11,025, DOCX 1340, XLS 1012, XLSX 240, PPT 5, PPTX 5, others 107 |

It is common knowledge that malicious samples tend to mutate over time. Therefore, a suitable detection model must be capable of adapting to this change. To evaluate the model's robustness, we obtained a batch of new malicious documents containing macros from VirusTotal. These documents were submitted after the publication of the previous dataset. All of these samples have been identified as malicious by more than 10 antivirus software in VirusTotal. We collected 2885 malicious samples between 25 February 2021 and 20 May 2022. More details about this dataset can be found in Table 2. Using the labeling information provided by Virustotal, as can be seen from the table, nearly half of the malicious samples are obfuscated. For convenience, we will refer to the first dataset as Dataset1 and the second dataset as Dataset2.

**Table 2.** New malicious samples collected from Virustotal.

| Label | Malicious |
|---|---|
| number | 2885 |
| Details of document type | doc 1537, docx 58, xls 1193, xlsx 16, ppt 13, others 68 |
| Details of obfuscation | Obfuscated 1358, non-obfuscated 1527 |

*4.2. Feature Extraction*

In the following paragraphs, we will explain in detail how we extract obfuscated features and which suspicious keywords we use.

4.2.1. Obfuscation Features

It is important to clarify that our focus is on detecting malicious macros, not obfuscated ones. While we list six categories of obfuscation techniques in Section 2, some of these obfuscations are often essential for attackers who do not want the malicious documents to be easily detected by antivirus software. For example, malicious commands can be obfuscated by split, encoding, or embedding obfuscation. If the attacker also wants to conceal suspicious method calls, call obfuscation is essential. To detect malicious macros, it is unnecessary and challenging to extract features from all forms of obfuscation. Instead, we focus on commonly used obfuscations by attackers, including split, encoding, and call obfuscation.

When extracting obfuscation features, the following principles are followed: (1) they should be clear and simple, requiring no complex analysis of macro semantics, and (2) they must be resistant to adversarial attacks. For instance, previous studies have considered the length of comments in macros as an obfuscation feature [2,20]. However, since adding comments to macros makes it easy for attackers to create adversarial samples, we argue that this feature lacks robustness. Therefore, we decided to remove comments before feature extraction. Symbols for convenience in understanding our proposed features are introduced in Table 3. The features and their descriptions are presented in Table 4.

**Table 3.** Symbols and their description.

| Symbols | Description |
|---|---|
| $L = \langle l_1, l_2, \ldots, l_n \rangle$ | $l_i$ represents a line in a macro's procedures; L is the vector of all lines in a macro. |
| $P = \langle p_1, p_2, \ldots, p_m \rangle$ | $p_i$ represents a procedure in a macro; P is the vector of all procedures in a macro. |
| count_concatenation | Count the number of concatenation symbols, including "+" and "&". |
| count_arithmetic | Count the number of arithmetic operators, including "+", "-", "*", and "/" |
| count_parentheses | Count the number of left parentheses adjacent to a word. |
| count_assignment | Count the number of " = ". |
| count_strings | Count the number of strings. |
| length | Calculate the length. |
| max | Calculate the maximum. |

**Table 4.** Features used to detect malicious macros.

| Feature Name | Description |
|---|---|
| F1 | $\max\{length(l_i)\}$, i from 1 to n |
| F2 | $\max\{count\_concatenation(l_i)\}$, i from 1 to n |
| F3 | $\max\{count\_arithmetic(l_i)\}$, i from 1 to n |
| F4 | $\max\{count\_parentheses(l_i)\}$, ifrom 1 to n |
| F5 | $\max\{count\_strings(l_i)\}$, i from 1 to n |
| F6 | $\max\{count\_concatenation(p_i)\}$, i from 1 tom |
| F7 | $\max\{count\_arithmetic(p_i)\}$, i from 1 tom |
| F8 | $\max\{count\_parentheses(p_i)\}$, i from 1 tom |
| F9 | $\max\{count\_strings(p_i)\}$, i from 1 to m |
| F10 | $\max\{count\_assignment(p_i)\}$, i from 1 to m |
| F11 | $length(P)$ |
| F12 | $length(L)$ |
| F13 | the frequency of function "CallByName" |
| F14 | the frequency of each conversion function, math function, string function |
| F15 | the frequency of each suspicious keyword |

From the perspective of attackers, they can choose to limit the obfuscated malicious code to a single line, a single procedure, or multiple procedures within a macro. Therefore, we need to extract obfuscation features for each of these three cases.

Detecting coding obfuscation and split obfuscation in a single line can be achieved through the evaluation of F1–F5 values, which indicate the complexity of the line. F1 represents the maximum line length, while F2 and F5 represent the maximum number of split operators and strings within a line. When split obfuscation is used, the values of F2 and F5 in malicious macros are more likely to be higher than those in benign ones. Therefore, we use F2 and F5 to identify split obfuscation. F3 represents the maximum number of arithmetic operators that can occur in a line. Figure 5 illustrates that arithmetic operations are sometimes used in coding obfuscation to conceal malicious intent. F4 represents the maximum number of left parentheses adjacent to a word within a line. The left parenthesis adjacent to a word indicates a procedure call. This feature represents the count of procedure calls in a line, including conversion functions, mathematical functions, and string functions used in coding obfuscation.

To identify obfuscation within a single procedure, the use of F1–F5 is ineffective when an attacker disassembles the obfuscated code line by line throughout the procedure. F6–F10 were implemented to address this issue. Features F6–F9 are similar to F2–F5, but their scope was changed from a line to a procedure. These features may reflect both split obfuscation and coding obfuscation within a procedure. Additionally, F10 identifies the maximum number of assignment operators ("=") present in procedures. If a line is broken down into multiple lines, more assignment operators will be used.

If an attacker spreads the obfuscated code across multiple procedures and tries to make each procedure's features F1–F10 appear as normal as possible, it will require more procedures and lines. This is comparable to adding a spoonful of salt to a large pool of water; more water will be needed to dilute the saltiness of the water. We use F11 and F12 as indicators of this type of dilution. F11 represents the number of procedures in a macro, while F12 represents the total number of lines in a macro.

In addition to the aforementioned statistical features, we also used several commonly employed functions for obfuscation. F13 illustrates the frequency of the "CallByName" function used in macros, which can provide insight into call obfuscation. F14 is a collection of sub-features that display the frequency of each conversion function, math function, and string function, such as "Asc", "Chr", "Mid", "Join", "InStr", "Replace", "Right", "StrConv", "Abs", "Atn", "Hex", and "Oct". In total, there are 64 functions included in F14. We use this feature to identify obfuscation which is statistically insignificant.

### 4.2.2. Suspicious Keyword Features

Malicious macros are typically more likely to use specific functions or strings to execute commands compared to normal ones. For instance, benign macros typically do not use the "Shell" function to execute programs, whereas attackers frequently employ this function. Table 5 indicates the frequently used keywords in malicious macros. F15 identifies 46 of these keywords as suspicious. It is worth mentioning that the selected keywords for detecting malicious macros are not comprehensive due to their complexity [21]. Only the most frequently used suspicious keywords were included.

**Table 5.** Features of suspicion to detect malicious macro.

| Keywords | Description |
|---|---|
| Auto_Open*, AutoOpen*, Document_Open*, Workbook_Open*, Document_Close* | Procedure names will be executed automatically upon opening or closing the document |
| CreateObject*, GetObject, Wscript.Shell*, Shell.Application* | Methods and parameters can be used to obtain the key object capable of executing commands |
| Shell*, Run*, Exec, Create, ShellExecute* | Methods can be used to execute a command or launch a program |
| CreateProcessA, CreateThread, CreateUserThread, VirtualAlloc, VirtualAllocEx, RtlMoveMemory, WriteProcessMemory, VirtualProtect, SetContextThread, QueueApcThread, WriteVirtualMemory, | External functions, when imported from kernel32.dll, can be used to create a process or thread, operating memory |
| Print*, FileCopy*, Open*, Write*, Output*, SaveToFile*, CreateTextFile*, Kill*, Binary* | Methods related to file creation, opening, writing, copying, deletion, etc. |
| cmd.exe, powershell.exe, vbhide* | Command line tools and suspicious parameters |
| StartupPath, Environ*, Windows*, ShowWindow*, dde*, Lib*, ExecuteExcel4Macro*, System*, Virtual* | Other keywords related to startup Path, environment variables, program windows, dde, function reference of DLL, execution of Excel 4 macro, virtualization |

Keywords with an asterisk are also used in [15].

## 5. Evaluation

After extracting the features from both Dataset1 and Dataset2, we trained four classical machine learning models to detect malicious macros. These models include random forest (RF), multi-layer perceptron (MLP), support vector machine (SVM), and K-nearest neighbor (KNN). The configuration parameters for each model are listed in Table 6. For the SVM and KNN models, their input was standardized before testing and training.

**Table 6.** Configuration of models.

| Model | Parameter |
|---|---|
| RF | n_estimators = 100 |
| MLP | hidden_layer_sizes = 150, max_iter = 500 |
| SVM | kernel = 'rbf' |
| KNN | n_neighbors = 3 |

We initially trained and validated the aforementioned models on Dataset1 using different feature selections. This includes obfuscation features (F1–F14), suspicious keyword features (F15), and combined features (F1–F15). Eighty percent of Dataset1 was used for training, and the remaining twenty percent was used for validation. Our experiments were conducted on a Huawei 2488V5 server with 2 Xeon (R) Gold 5118 CPUs, 64 GB system memory, and the CentOS 8 Stream operating system. We evaluated the performance of the trained classifiers using standard classification metrics, including precision, recall, accuracy, and F1-score. Considering the imbalance of malicious and benign samples in Dataset1, we also incorporated the false alarm rate (FAR) into the metrics.

Table 7 displays the performance of various models obtained through 5-fold cross-validation on Dataset1. To compare with the work presented in [15], we added the best results reported in their study to the final row of Table 7.

**Table 7.** Performance results of models on Dataset1.

| Model | Feature Selection | FAR | Precision | Recall | Accuracy | F1-Score |
|---|---|---|---|---|---|---|
| | F1–F14 | 0.083 | 0.982 | 0.994 | 0.981 | 0.988 |
| RF | F15 | 0.012 | 0.997 | 0.995 | 0.993 | 0.996 |
| | F1–F15 | 0.015 | 0.997 | 0.997 | 0.994 | 0.997 |
| | F1–F14 | 0.138 | 0.971 | 0.979 | 0.958 | 0.975 |
| MLP | F15 | 0.013 | 0.997 | 0.995 | 0.994 | 0.996 |
| | F1–F15 | 0.035 | 0.993 | 0.994 | 0.989 | 0.993 |
| | F1–F14 | 0.203 | 0.957 | 0.961 | 0.932 | 0.959 |
| SVM | F15 | 0.022 | 0.995 | 0.987 | 0.986 | 0.991 |
| | F1–F15 | 0.028 | 0.994 | 0.992 | 0.988 | 0.993 |
| | F1–F14 | 0.092 | 0.980 | 0.984 | 0.971 | 0.982 |
| KNN | F15 | 0.020 | 0.996 | 0.992 | 0.990 | 0.994 |
| | F1–F15 | 0.028 | 0.994 | 0.992 | 0.988 | 0.993 |
| RF | Ref. [15] | — | 0.993 | 0.976 | 0.975 | 0.985 |

After conducting cross-validation experiments on Dataset1, the entire dataset was used to train our four machine learning classifiers. Subsequently, we conducted the same experiment on Dataset2, which consisted solely of malicious samples. This time, we used precision as the metric for Dataset2. Table 8 shows the results.

From Table 7, it is evident that the RF classifier with features F1–F15 achieves the highest F1-score among all models. Furthermore, classifiers using suspicious keywords (F15) always yield better detection performance than obfuscation features (F1–F14). At first glance, incorporating obfuscation features that rely on suspicious keywords does not appear to have a significant impact on detection performance and may even slightly worsen it for MLP and KNN models. Nonetheless, Table 8 clearly shows that models that rely solely on suspicious keywords as features experience a significant decline in performance. Interestingly, however, when considering the combined features, the random forest model achieves a precision of 0.953 on the 2885 new malicious samples. This indicates that integrating obfuscation features and suspicious keywords enhances the resilience of the model.

**Table 8.** Performance results of models on Dataset2.

| Model | Feature Selection | Precision |
|---|---|---|
| RF | F1–F14 | 0.923 |
| | F15 | 0.856 |
| | F1–F15 | **0.953** |
| MLP | F1–F14 | 0.739 |
| | F15 | 0.793 |
| | F1–F15 | **0.907** |
| SVM | F1–F14 | 0.799 |
| | F15 | 0.788 |
| | F1–F15 | **0.817** |
| KNN | F1–F14 | 0.792 |
| | F15 | 0.757 |
| | F1–F15 | **0.735** |

Compared with the work in [15], our RF classifier trained using suspicious keywords (F15) has a higher recall and F1-score. This is likely due to the inclusion of new suspicious keywords, such as "GetObject" and external functions, in our features. These additions enable us to detect more malicious samples.

Table 9 displays the top 20 features of the RF classifier, which use combined features to identify malicious samples in Dataset2. It reveals that only half of the prominent features consist of suspicious keywords. Additionally, ten obfuscation features, including four statistical features, demonstrate their significance in detecting malicious macros. This highlights the potential for complementarity between both types of features in improving detection performance. We further analyzed the proportion of benign and malicious samples containing keywords from F14 and F15. Table 10 demonstrates that samples containing these keywords are more likely to be malicious.

**Table 9.** The top 20 prominent features and their type.

| Index | Feature Name | Feature Type |
|---|---|---|
| 1 | F15:CreateObject | Suspicious Keywords |
| 2 | F15:Document_Open | Suspicious Keywords |
| 3 | F15:Shell | Suspicious Keywords |
| 4 | F15:GetObject | Suspicious Keywords |
| 5 | F15:Lib | Suspicious Keywords |
| 6 | F15:AutoOpen | Suspicious Keywords |
| 7 | F15:Auto_Open | Suspicious Keywords |
| 8 | F15:StartupPath | Suspicious Keywords |
| 9 | F11:length(P) | Obfuscation |
| 10 | $F3:\max\{count\_arithmetic(l_i)\}$ | Obfuscation |
| 11 | F14:Chr | Obfuscation |
| 12 | F14:Asc | Obfuscation |
| 13 | F14:UCase | Obfuscation |
| 14 | $F7:\max\{count\_arithmetic(p_i)\}$ | Obfuscation |
| 15 | $F5:\max\{count\_strings(l_i)\}$ | Obfuscation |
| 16 | F14:Left | Obfuscation |
| 17 | F15:Open | Suspicious Keywords |
| 18 | F14:Abs | Obfuscation |
| 19 | F14:Split | Obfuscation |
| 20 | F15:System | Suspicious Keywords |

**Table 10.** Proportion of benign and malicious for samples containing keywords from F14 and F15.

| Index | Feature Name | Proportion of Benign Samples | Proportion of Malicious Samples |
|---|---|---|---|
| 1 | F15:CreateObject | 2.76% | 97.24% |
| 2 | F15:Document_Open | 0.41% | 99.59% |
| 3 | F15:Shell | 0.29% | 99.71% |
| 4 | F15:GetObject | 0.07% | 99.93% |
| 5 | F15:Lib | 1.72% | 98.28% |
| 6 | F15:AutoOpen | 0.12% | 99.88% |
| 7 | F15:Auto_Open | 9.62% | 90.38% |
| 8 | F15:StartupPath | 0.00% | 100.00% |
| 9 | F14:Chr | 2.31% | 97.69% |
| 10 | F14:Asc | 1.88% | 98.12% |
| 11 | F14:UCase | 34.20% | 65.80% |
| 12 | F14:Left | 10.94% | 89.06% |
| 13 | F15:Open | 5.75% | 94.25% |
| 14 | F14:Abs | 27.32% | 72.68% |
| 15 | F14:Split | 3.45% | 96.55% |
| 16 | F15:System | 12.27% | 87.73% |

We conducted further investigation to understand why there are still some samples in Dataset2 that cannot be detected by the RF classifier when using combined features. It is interesting to note that some of these undetected samples can be identified by the RF classifier using either feature selection F1–F14 or F15 alone. A schematic diagram is included to illustrate sample detection using the RF classifier on Dataset2 (see Figure 9). The samples successfully detected by the corresponding classifier are represented by the points inside the circle, while the points outside the circle depict the undetected samples. It is apparent that each classifier can identify a subset of the samples. We believe that the combination of features may weaken certain critical characteristics that may appear obvious and easy to learn from only one type of feature. These features are then placed at lower-level nodes in many of the decision trees within the random forest. This may affect the classification results for certain samples that are accurately classified based on a single type of feature.

Based on the above experiments and analyses, we can improve our detection capability by integrating multiple classifiers. Table 11 illustrates the precision of Dataset2 using different ensembles of classifiers. If we combined the RF classifier with F15 and the RF classifier with F1–F15, the precision would be 0.969. When we combine all three RF classifiers to detect samples in Dataset2, we achieve a precision of 0.980. In addition, most of the undetected samples are malicious macros based on P-Code, which is beyond the scope of this paper. It should be noted that this simple combination may lead to an increase in false positives. Table 6 shows that the classifier with features F1–F14 consistently has a higher rate of false positives compared to the other classifiers. Therefore, it depends on specific scenarios when combining these classifiers.

**Table 11.** Precision with different ensembles of classifiers.

| Ensemble of Classifier | Precision |
|---|---|
| RF classifier with F1–F14 and RF classifier with F15 | 0.979 |
| RF classifier with F1–F14 and RF classifier with F1–F15 | 0.974 |
| RF classifier with F15 and RF classifier with F1–F15 | 0.969 |
| All three classifiers | 0.980 |

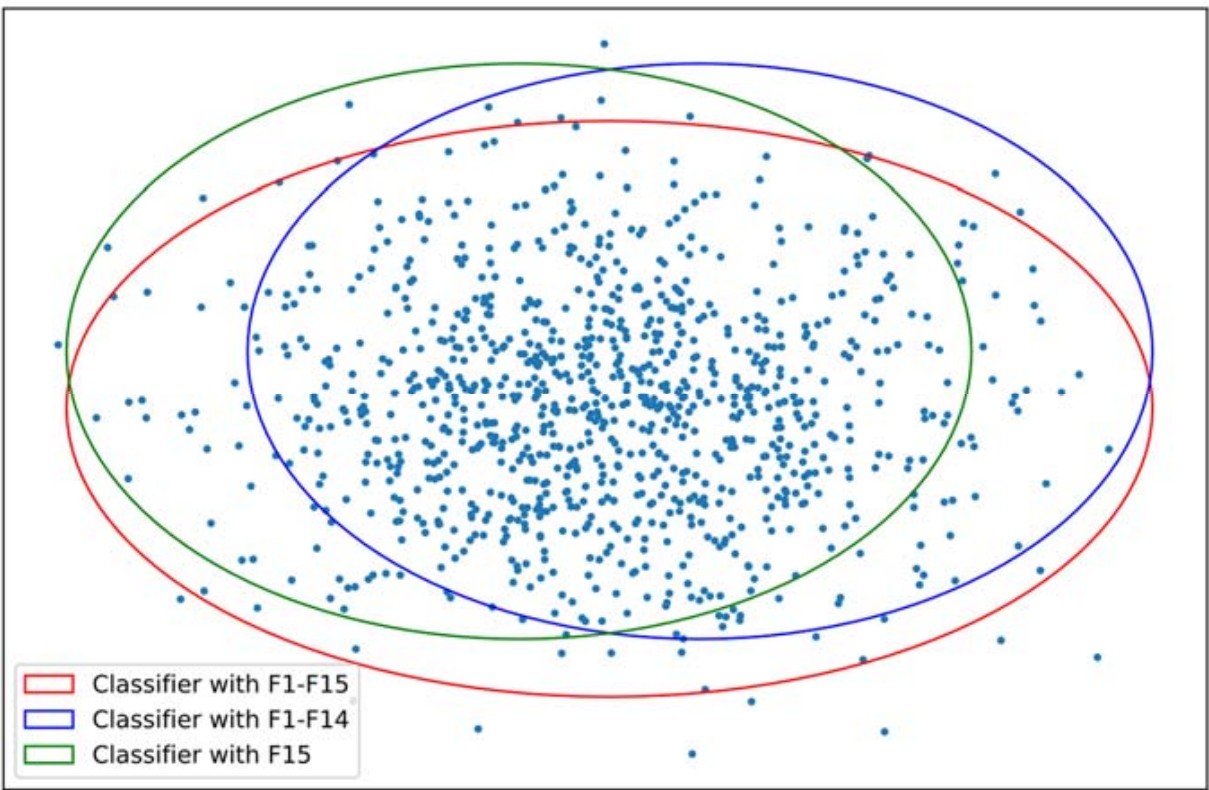

**Figure 9.** Sample detection diagram of RF classifier.

## 6. Discussion

We analyze both obfuscation features and suspicious keywords to identify malicious macros. We assume that malicious macros are always obfuscated and that specific functions or strings are essential components. This enables our model to identify malicious samples using keywords that are previously unknown to the community. If experts can identify these new keywords from the samples and then use them to retrain the model with additional features and samples, similar to the approach taken in [14], the model will maintain a higher level of robustness. However, this does not mean that attackers cannot bypass our model. Currently, our obfuscation features do not include embedding obfuscation. This is because it is difficult to distinguish embedding obfuscation between benign macros and malicious macros. If an attacker combines embedding obfuscation with newly discovered attack techniques, it is highly likely that our classifier will be bypassed. This issue remains to be resolved in our future work.

## 7. Conclusions

In this paper, we propose a novel approach to enhance the detection of malicious macros by combining obfuscation features and suspicious keywords. We extract 78 obfuscation features and 46 suspicious keywords. Experiments are conducted on two datasets: Dataset1, which consists of 2939 benign samples and 13,734 malicious samples, and Dataset2, which contains 2885 new samples that were reported after the publication date of Dataset1. The initial experiment conducted on Dataset1 indicates that the highest performing classifier is random forest (RF) with combined features, achieving an F1-score of 0.997. When using the model trained on Dataset1 to identify samples in Dataset2, the precision of the RF classifier with suspicious keywords decreases significantly from 0.996 to 0.856. In contrast, the RF classifier with combined features is capable of detecting 95.3% of malicious samples. This demonstrates that the combined features can maintain model robustness while achieving a high detection rate. The analysis of the top 20 prominent features reveals

that both obfuscated features and suspicious keywords play a crucial role in identifying malicious macros.

**Author Contributions:** Writing—original draft, X.C.; Writing—review & editing, W.W.; Supervision, W.H. All authors have read and agreed to the published version of the manuscript.

**Funding:** This work is support by National Natural Science Foundation of China (Project No. 62176264).

**Institutional Review Board Statement:** Not applicable.

**Informed Consent Statement:** Not applicable.

**Data Availability Statement:** The data and source code presented in this study are available at https://gitee.com/chensheng101/macro_fc (accessd on 18 September 2023).

**Conflicts of Interest:** The authors declare no conflict of interest.

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
