# Peer review of "Malicious Office Macro Detection: Combined Features with Obfuscation and Suspicious Keywords"

_applsci, doi:10.3390/app132212101_

Round 1

Reviewer 1 Report

Comments and Suggestions for Authors

In this paper, the authors analyze 77 obfuscation features from the attacker's perspective and extract 46 suspicious keywords in the macro. They first combined the above two types of features to train machine learning models on the public dataset. Then the same experiment on a dataset consisting of a collection of newly discovered samples to see whether their proposed method could discover unseen malicious macros. Experimental results show that compared with existing research, their proposed method has a higher detection rate and better consistency. Moreover, combining multiple classifiers with different feature selection further improves the detection performance.

However, in order to improve the academic aspect of the article, the following issues need to be corrected.

1) It would be more academic to provide the pseudo codes of the code pieces in the article.

2) Flow diagram should be used in appropriate blocks.

3) References are not academic. The URL links provided should be given in accordance with the MDPI format, and non-academic links should be removed.

4) The number of references and literature review should be enriched.

5) Main and subheadings should not be consecutive. For example, after the title of Chapter 2, a text covering the subject should be written, even if it is 1 paragraph, and then proceed to the title 2.1. This situation should be corrected in other sections as well. (Ex: Chapter 4)

6) Codes cannot be given as Figures. Either the Code must be given as a list or it must be given as pseudocode.

7) The Related Work section is kept very short and insufficient.

8) There is no need for a Discussion section. The discussion part in that section can be given in the application results section.

9) The "Conclusion" section must be independent.

Comments on the Quality of English Language

Suitable. 

Author Response

Thank you very much for taking the time to review this manuscript. Please find the detailed responses below.

1.Point-by-point response to Comments and Suggestions for Authors

1) It would be more academic to provide the pseudo codes of the code pieces in the article.

Response :We need to use real code in order to show how macro obfuscation is achieved, which it's impossible with pseudo-codes

2) Flow diagram should be used in appropriate blocks.

Response :Agree, we will include a flow diagram to describe how our model works. This can improve the reader's understanding of our proposed method.

3) References are not academic. The URL links provided should be given in accordance with the MDPI format, and non-academic links should be removed.

Response: Agree. We will remove non-academic links and modify the references with MDPI format.

4) The number of references and literature review should be enriched.

Response:  Agree. We will enrich the references with more researches about maldoc’s detection, not just about malicious macro’s detection.

5) Main and subheadings should not be consecutive. For example, after the title of Chapter 2, a text covering the subject should be written, even if it is 1 paragraph, and then proceed to the title 2.1. This situation should be corrected in other sections as well. (Ex: Chapter 4)

Response: Agree. We will add a descriptive paragraph after each main heading to increase the readability of the article.

6) Codes cannot be given as Figures. Either the Code must be given as a list or it must be given as pseudocode.

Response: Actually, some related papers also use figures to describe macro obfuscation, such as paper Obfuscated VBA macro detection using machine learning.

7) The Related Work section is kept very short and insufficient.

Response: This is mainly due to the fact that there are not many studies dedicated to the detection of malicious macros in documents, in order to enrich the related research work, we will add some related works on malicious document detection.

8) There is no need for a Discussion section. The discussion part in that section can be given in the application results section.

Response: We think it may not be appropriate to move the discussion part to application results section, so we put it in discussion section.

9) The "Conclusion" section must be independent.

Response: After the remove of discussion in the final section, the "Conclusion" section will be independent.

2. Response to Comments on the Quality of English Language

1)Extensive editing of English language required

Response: We will have our manuscript checked by a native English-speaking colleague.

Reviewer 2 Report

Comments and Suggestions for Authors

Dear authors,

I checked your manuscript in detail and have several comments. Please, take them into account during the improvement of the manuscript.

Comment 1. The description of the dataset used shows the prevalence of the malicious documents (13734) on benign documents (2939). Moreover, some document types are present only as malicious ones (for example, ppt 5, pptx 5, and others 107). It feels like more representative dataset is required, where the difference between malicious and benign documents is not easy to find.

Comment 2. The manuscript contains the comparative analysis of performance of different machine learning methods in malicious office macros detection. And while the authors compared results of different methods, the comparison with results of other researchers is missing.

Comment 3. Most of the references are not scientific papers, but links to different news websites and blogs. It might be better to put such references as footnotes, while extend the related work section with additional scientific papers. 

Author Response

Thank you very much for taking the time to review this manuscript. Please find the detailed responses below.

Comment 1. The description of the dataset used shows the prevalence of the malicious documents (13734) on benign documents (2939). Moreover, some document types are present only as malicious ones (for example, ppt 5, pptx 5, and others 107). It feels like more representative dataset is required, where the difference between malicious and benign documents is not easy to find.

Response 1. Benign documents with macros usually involve private data of companies and are more difficult to obtain from the Internet than with malicious documents, therefore, we directly use publicly available datasets. Though some document types are present only as malicious ones (for example, ppt 5, pptx 5, and others 107), the VBA macros in these type of documents have no difference.

Comment 2. The manuscript contains the comparative analysis of performance of different machine learning methods in malicious office macros detection. And while the authors compared results of different methods, the comparison with results of other researchers is missing.

Response 2. Actually we compared our method with one previous research in table 7, the results on the same dataset shows our method archived better performance. Although there are some other related studies, they use their own datasets and comparing performance results on different datasets is not convincing, so we have not put their results for comparison here.

Comment 3. Most of the references are not scientific papers, but links to different news websites and blogs. It might be better to put such references as footnotes, while extend the related work section with additional scientific papers.

Response 3. This is mainly due to the fact that there are not many studies dedicated to the detection of malicious macros in documents, in order to enrich the related research work, we will add some literature on malicious document detection.

Reviewer 3 Report

Comments and Suggestions for Authors

1)      Authors should add motivation and contribution in the introduction section

2)      How the proposed model is beneficial in decentralized system?

3)      Authors should give some discussion on consensus approach of mining the healthcare records.

4)      The grammar and typos error must be taken care 

5)      Author should add advantages and disadvantages of the proposed model.

6)      Is the proposed system secure enough and sustainable to apply in distributed environment? Justify it by giving some security parameters used in the model.

7)      How the model is providing privacy while making communication that must be detailed by the authors.

8)      Is the developed model secure in terms of different entities communication in the model? How the entity is secured in the model and how the confidentiality of the transactions are done?

9)      Authors should compare the work with existing work of healthcare using blockchain

a.       Blockchain-Based Authentication and Explainable AI for Securing Consumer IoT Applications." IEEE Transactions on Consumer Electronics 

Comments on the Quality of English Language

Minor spell check is required

Author Response

Thank you very much for taking the time to review this manuscript. Please find the detailed responses below.

1.Point-by-point response to Comments and Suggestions for Authors

1)Authors should add motivation and contribution in the introduction section

Response: We have described motivation and contribution in the introduction section.

2)How the proposed model is beneficial in decentralized system?

Response: Not related to our study.

3)Authors should give some discussion on consensus approach of mining the healthcare records.

Response: Not related to our study.

4)The grammar and typos error must be taken care 

Response: We will thoroughly check and revise the manuscript for grammatical and spelling errors

5)Author should add advantages and disadvantages of the proposed model.

Reponse: Actually we describe the disadvantages of our model in the final section.

6)Is the proposed system secure enough and sustainable to apply in distributed environment? Justify it by giving some security parameters used in the model.

Response: Not related to our study.

8)Is the developed model secure in terms of different entities communication in the model? How the entity is secured in the model and how the confidentiality of the transactions are done?

Response: Not related to our study.

9)Authors should compare the work with existing work of healthcare using blockchain

Response: Not related to our study.

2. Response to Comments on the Quality of English Language

1)Minor editing of English language required

Response: We will have our manuscript checked by a native English-speaking colleague

Reviewer 4 Report

Comments and Suggestions for Authors

The manuscript is about dealing with obfuscation techniques used by attackers to bypass Microsoft security controls. The authors created different models based on the extracted obfuscation features.

The manuscript is well organized and contains the different required validation materials. The proposed models might help in detecting obfuscation macros.

Comments on the Quality of English Language

Minor editing of English language required

Author Response

Thank you very much for taking the time to review this manuscript. In response to your comment  about the quality of English language, we will have our manuscript checked by a native English-speaking colleague.

Round 2

Reviewer 1 Report

Comments and Suggestions for Authors

The authors took my comments into consideration and the necessary corrections were addressed in the article.